# Derivation of Equivalent Material Coefficients of 2-2 Piezoelectric Single Crystal Composite

**DOI:** 10.3390/mi15070917

**Published:** 2024-07-16

**Authors:** Minseop Sim, Yub Je, Yohan Cho, Hee-Seon Seo, Moo-Joon Kim

**Affiliations:** 1Agency for Defense Development, Jinhae P.O. Box 18, Changwon 51678, Republic of Korea; simms@add.re.kr (M.S.); sideze@add.re.kr (Y.J.); yhcho@add.re.kr (Y.C.); hsseo@add.re.kr (H.-S.S.); 2Department of Physics, Pukyong National University 45, Yongso-ro, Nam-gu, Busan 48513, Republic of Korea

**Keywords:** 2-2 piezoelectric composite, single crystals, PIN-PMN-PT, analytic solution, equivalent material coefficients

## Abstract

Piezoelectric composites, which consist of piezoelectric materials and polymers, are widely employed in various applications such as underwater sonar transducers and medical diagnostic ultrasonic transducers. Acoustic transducers based on piezoelectric composites can have high sensitivity with broad bandwidth. In recent studies, it is demonstrated that 2-2 composites based on single crystals provide further increased sensitivity and wide bandwidth. In order to utilize a 2-2 composite in acoustic sensors, it is required to demonstrate the full material coefficients of the 2-2 composite. In this study, we investigated an analytic solution for determining equivalent material coefficients of a 2-2 composite. Impedance spectrums of the single-phase resonators with equivalent material coefficients and 2-2 composite resonators were compared by the finite element method in order to verify the analytic solutions. Furthermore, the equivalent material coefficients derived from the analytic solution were also verified by comparing the measured and the simulated impedance spectrums. The difference in resonance and anti-resonance frequencies between the measured and simulated impedance spectrums was around 0.5% and 1.2%. By utilizing the analytic solutions in this study, it is possible to accurately derive full equivalent material coefficients of a 2-2 composite, which are essential for the development of acoustic sensors.

## 1. Introduction

Piezoelectric composites, which consist of piezoelectric materials and polymers, are widely employed in various applications such as underwater sonar transducers, medical diagnostic ultrasonic transducers, and non-destructive testing devices [1,2,3]. Piezoelectric composites have the principal advantage of being able to control material properties such as density, stiffness, and piezoelectric properties by changing the volume fractions of piezoelectric materials [4,5]. Therefore, acoustic transducers based on piezoelectric composites can provide wide bandwidth and high sensitivity by reducing the acoustic impedance of composites and by increasing piezoelectric coefficients of composites, respectively. There are various configurations of piezoelectric composites with i-j configuration, where ‘i’ represents the connectivity of the piezoelectric materials, and ‘j’ represents the connectivity of the polymer. Among them, piezoelectric composites with 1-3 connectivity and 2-2 connectivity are mostly used due to their high electro-mechanical coupling and simplicity of fabrication [6,7,8,9,10]. 

PbZrTiO_3_ (PZT) has been the most widely used active material for piezoelectric composites due to its high piezoelectric properties and electro-mechanical coupling factors. Since PZT ceramic composites can be cost-effectively fabricated by the injection molding process, acoustic sensors based on PZT ceramic composites have been successfully commercialized [5]. There has been increasing interests for acoustic sensors based on composites with further increased sensitivity and bandwidth by changing the active material of composites [11,12,13]. In recent research [14,15,16,17,18,19,20], ferroelectric piezoelectric single crystal materials such as Pb(Mg_1/3_Nb_2/3_)O_3_-PbTiO_3_(PMN-PT) and Pb(Zn_1/3_Nb_2/3_)O_3_-PbTiO_3_ (PZN-PT) have been investigated as active materials in piezoelectric composites. Since single crystals provide superior piezoelectric coefficients (d_33_ ~1500 pC/N) and electromechanical coupling coefficients (k_33_~0.9) compared to PZT ceramics, acoustic sensors based on single crystal composites can increase their sensitivity and bandwidth [19,20,21,22].

Single crystal composites with 1-3 connectivity have been primarily investigated for acoustic sensors, but there are difficulties in their fabrication process in terms of the dice and fill process. On the other hand, single crystal composites with 2-2 connectivity have advantages in terms of the dice and fill fabrication process due to their configuration. Lili et al. [17] reported that 2-2 single crystal composites, which are lamellar stacks of [011] poled PMN-PT single crystal layers and polymer layers, show a superior hydrostatic figure of merit (FOM) over 1-3 single crystal composites. In our previous study [23], it is shown that 2-2 composites based on [011] poled PIN-PMN-PT single crystals can have higher sensitivity and wider bandwidth compared to piezoelectric composites with other configurations when they are used for hydrophones with a thickness resonance mode [24,25].

Piezoelectric composites with 2-2 connectivity exhibit anisotropic properties because of their configurations and the anisotropic properties of piezoelectric materials, whereby [011] poled PIN-PMN-PT single crystals, having a macroscopic orthorhombic structure with mm2 symmetry, have a total of seventeen independent material coefficients, consisting of nine elastic coefficients, five5 piezoelectric coefficients, and three permittivity coefficients. The 2-2 composite, consisting of [011] poled PIN-PMN-PT single crystals and polymers, also exhibits the macroscopic mm2 symmetry with 17 independent material coefficients. Therefore, it is essential to identify the full material coefficients of the 2-2 composites based on single crystals for applications in acoustic sensors [26,27,28,29].

To identify the material properties of piezoelectric composites, static testing is typically employed [27,30,31]. Static testing provides their mechanical and electrical properties under static or quasi-static loading conditions. While static testing makes it easy to measure the material properties in the primary direction, identifying the full anisotropic material properties of piezoelectric composites is rather complicated. Numerical simulation, based on the finite element method, can be another way to identify the material properties of piezoelectric composites [32,33,34]. The material properties of piezoelectric composites can be obtained from simulated impedance curves of the resonators. This method, based on numerical simulation, is particularly effective at handling complex geometries, boundary conditions, and anisotropic materials. However, numerical simulation requires considerable simulation time and also requires re-simulation of the material properties if there are any changes in the volume fraction or configuration of the piezoelectric composite. Theoretical analytic solutions, derived by constitutive equations and boundary conditions, provide a significant advantage for identifying the material properties of piezoelectric composites. Using analytic solutions, the material properties of piezoelectric composites can be easily obtained without rather complex simulations or repetitive testing. In previous studies [35,36], analytic solutions for 1-3 or 2-2 piezoelectric composites have investigated based on PZT ceramics.

There have been the several studies on theoretical models of a 2-2 composite based on single crystals. Most of the previous studies investigated the theoretical model only for the primary material properties that are closely related to the transducer’s performance such as d_h_, g_h_, and figure of merit (FOM) [37,38,39]. In other previous studies, the sound velocity, electrical impedance, sensitivity, and bandwidth of transducers based on 2-2 single crystal composites have been theoretically and experimentally characterized [40,41]. However, it is required to derive a theoretical model for all 17 material coefficients of 2-2 single crystal composites for detailed theoretical and numerical analysis when designing acoustic sensors based on 2-2 single crystal composites.

Therefore, in this study, we investigate analytic solutions for the equivalent material properties of a 2-2 composite based on [011] poled single crystals. The analytic solutions were derived by considering constitutive equations and the corresponding boundary conditions of the 2-2 composite. The analytic solution and the derived equivalent material properties of the 2-2 composite are verified by finite element analysis. To verify 17 material coefficients of the 2-2 composites of the [011] poled PIN-PMN-PT single crystals and polymers, eight resonators are introduced. Each of these resonators is designed to predominantly vibrate in their principal directions, corresponding to specific vibrational modes. The impedance spectrums of single-phase resonators with the derived equivalent material properties are compared with the impedance spectrum of fully modeled resonators with a 2-2 configuration. A 2-2 composite utilizing [011] poled PIN-PMN-PT and Epotek 301 epoxy is fabricated. The measured impedance spectrum is also compared to the simulated impedance spectrum based on equivalent material properties in order to verify the analytic solutions. 

The background and motivation for this research are described in Section 1. The derivation of the analytic solutions of the equivalent material coefficients is described in Section 2. The finite element analysis and experiments are described in Section 3 and Section 4. The detailed derivation of the analytic solutions of the equivalent material coefficients of the 2-2 composites are described in Appendix A, respectively.

## 2. Analytic Solution of Equivalent Material Coefficients

We investigate the analytic solution of equivalent material coefficients of the 2-2 composite consisting of [011] poled single crystals and polymers. Figure 1 shows the 2-2 composites, which are lamellar stacks of the [011] poled single crystal layers and the polymer layers. The [011] poled relaxor-PT single crystals such as Pb(Mg_1/3_Nb_2/3_)O_3_-PbTiO_3_ (PMN-PT) and Pb(Zn_1/3_Nb_2/3_)O_3_-PbTiO_3_ (PZN-PT) exhibit positive piezoelectric coefficients in the 1 and 3 directions, while they have negative piezoelectric coefficients in the 2 direction. The 2-2 composites have an advantage in their configuration since the polymer layers stacked along the 2 direction reduce the destructive interference from the negative piezoelectric coefficient in the 2 direction.

The [011] poled single crystals exhibit a macroscopic orthorhombic structure with mm2 symmetry, which has seventeen independent material coefficients: nine elastic coefficients, five piezoelectric coefficients, and three permittivity coefficients. Polymers exhibit isotropic material coefficients, which have four independent material coefficients: three elastic coefficients and one permittivity coefficient. Composites of a 2-2 structure consisting of [011] poled single crystals and polymers also exhibit a macroscopic orthorhombic structure with mm2 symmetry, which has 17 independent material coefficients.

The constitutive equations of the [011] poled single crystals are as follows:(1a)$$\left(\begin{array}{c} T_{1}^{SC}\\ T_{2}^{SC}\\ T_{3}^{SC}\\ T_{4}^{SC}\\ T_{5}^{SC}\\ T_{6}^{SC} \end{array}\right)=\left(\begin{array}{cccccc} c_{11}^{SC} & c_{12}^{SC} & c_{13}^{SC} & 0 & 0 & 0\\ c_{12}^{SC} & c_{22}^{SC} & c_{23}^{SC} & 0 & 0 & 0\\ c_{13}^{SC} & c_{23}^{SC} & c_{33}^{SC} & 0 & 0 & 0\\ 0 & 0 & 0 & c_{44}^{SC} & 0 & 0\\ 0 & 0 & 0 & 0 & c_{55}^{SC} & 0\\ 0 & 0 & 0 & 0 & 0 & c_{66}^{SC} \end{array}\right)\left(\begin{array}{c} S_{1}^{SC}\\ S_{2}^{SC}\\ S_{3}^{SC}\\ S_{4}^{SC}\\ S_{5}^{SC}\\ S_{6}^{SC} \end{array}\right)-\left(\begin{array}{ccc} 0 & 0 & e_{13}^{SC}\\ 0 & 0 & e_{23}^{SC}\\ 0 & 0 & e_{33}^{SC}\\ 0 & e_{24}^{SC} & 0\\ e_{15}^{SC} & 0 & 0\\ 0 & 0 & 0 \end{array}\right)\left(\begin{array}{c} E_{1}^{SC}\\ E_{2}^{SC}\\ E_{3}^{SC} \end{array}\right)$$
(1b)$$\left(\begin{array}{c} D_{1}^{SC}\\ D_{2}^{SC}\\ D_{3}^{SC} \end{array}\right)=\left(\begin{array}{ccc} 0 & 0 & e_{13}^{SC}\\ 0 & 0 & e_{23}^{SC}\\ 0 & 0 & e_{33}^{SC}\\ 0 & e_{24}^{SC} & 0\\ e_{15}^{SC} & 0 & 0\\ 0 & 0 & 0 \end{array}\right)\left(\begin{array}{c} S_{1}^{SC}\\ S_{2}^{SC}\\ S_{3}^{SC} \end{array}\right)+\left(\begin{array}{ccc} \varepsilon_{11}^{SC} & 0 & 0\\ 0 & \varepsilon_{22}^{SC} & 0\\ 0 & 0 & \varepsilon_{33}^{SC} \end{array}\right)\left(\begin{array}{c} E_{1}^{SC}\\ E_{2}^{SC}\\ E_{3}^{SC} \end{array}\right)$$
where $T_{i}$ is a stress matrix, $S_{i}$ is a strain matrix, $D_{i}$ is an electric displacement matrix, and $E_{i}$ is an electric field matrix. $c_{ij}$ is a stiffness coefficient matrix, $e_{ij}$ is a piezoelectric coefficient matrix, and $\varepsilon_{ij}$ is a permittivity coefficient matrix. The superscript *SC* indicates single crystals.

Similarly, the constitutive equations of polymer are as follows:(1c)$$\left(\begin{array}{c} T_{1}^{P}\\ T_{2}^{P}\\ T_{3}^{P}\\ T_{4}^{P}\\ T_{5}^{P}\\ T_{6}^{P} \end{array}\right)=\left(\begin{array}{cccccc} c_{11}^{P} & c_{12}^{P} & c_{12}^{P} & 0 & 0 & 0\\ c_{12}^{P} & c_{11}^{P} & c_{12}^{P} & 0 & 0 & 0\\ c_{12}^{P} & c_{12}^{P} & c_{11}^{P} & 0 & 0 & 0\\ 0 & 0 & 0 & c_{44}^{P} & 0 & 0\\ 0 & 0 & 0 & 0 & c_{44}^{P} & 0\\ 0 & 0 & 0 & 0 & 0 & c_{44}^{P} \end{array}\right)\left(\begin{array}{c} S_{1}^{P}\\ S_{2}^{P}\\ S_{3}^{P}\\ S_{4}^{P}\\ S_{5}^{P}\\ S_{6}^{P} \end{array}\right)$$
(1d)$$\left(\begin{array}{c} D_{1}^{P}\\ D_{2}^{P}\\ D_{3}^{P} \end{array}\right)=\left(\begin{array}{ccc} \varepsilon_{11}^{P} & 0 & 0\\ 0 & \varepsilon_{11}^{P} & 0\\ 0 & 0 & \varepsilon_{11}^{P} \end{array}\right)\left(\begin{array}{c} E_{1}^{P}\\ E_{2}^{P}\\ E_{3}^{P} \end{array}\right)$$
where superscript *P* indicates polymer.

The constitutive equations of 2-2 composites with mm2 symmetry are as follows:(1e)$$\left(\begin{array}{c} T_{1}^{C}\\ T_{2}^{C}\\ T_{3}^{C}\\ T_{4}^{C}\\ T_{5}^{C}\\ T_{6}^{C} \end{array}\right)=\left(\begin{array}{cccccc} c_{11}^{C} & c_{12}^{C} & c_{13}^{C} & 0 & 0 & 0\\ c_{12}^{C} & c_{22}^{C} & c_{23}^{C} & 0 & 0 & 0\\ c_{13}^{C} & c_{23}^{C} & c_{33}^{C} & 0 & 0 & 0\\ 0 & 0 & 0 & c_{44}^{C} & 0 & 0\\ 0 & 0 & 0 & 0 & c_{55}^{C} & 0\\ 0 & 0 & 0 & 0 & 0 & c_{66}^{C} \end{array}\right)\left(\begin{array}{c} S_{1}^{C}\\ S_{2}^{C}\\ S_{3}^{C}\\ S_{4}^{C}\\ S_{5}^{C}\\ S_{6}^{C} \end{array}\right)-\left(\begin{array}{ccc} 0 & 0 & e_{13}^{C}\\ 0 & 0 & e_{23}^{C}\\ 0 & 0 & e_{33}^{C}\\ 0 & e_{24}^{C} & 0\\ e_{15}^{C} & 0 & 0\\ 0 & 0 & 0 \end{array}\right)\left(\begin{array}{c} E_{1}^{C}\\ E_{2}^{C}\\ E_{3}^{C} \end{array}\right)$$
(1f)$$\left(\begin{array}{c} D_{1}^{C}\\ D_{2}^{C}\\ D_{3}^{C} \end{array}\right)=\left(\begin{array}{ccc} 0 & 0 & e_{13}^{C}\\ 0 & 0 & e_{23}^{C}\\ 0 & 0 & e_{33}^{C}\\ 0 & e_{24}^{C} & 0\\ e_{15}^{C} & 0 & 0\\ 0 & 0 & 0 \end{array}\right)\left(\begin{array}{c} S_{1}^{C}\\ S_{2}^{C}\\ S_{3}^{C} \end{array}\right)+\left(\begin{array}{ccc} \varepsilon_{11}^{C} & 0 & 0\\ 0 & \varepsilon_{22}^{C} & 0\\ 0 & 0 & \varepsilon_{33}^{C} \end{array}\right)\left(\begin{array}{c} E_{1}^{C}\\ E_{2}^{C}\\ E_{3}^{C} \end{array}\right)$$
where the superscript *C* indicates the composite.

In order to derive the analytic solution of the equivalent material coefficients of the 2-2 composites, their boundary conditions are established. In the 2-2 configuration, the single crystals and polymers exhibit identical strain along the 1, 3, 4, and 5 directions, as shown in Equation (2a). On the other hand, the total stress on the 2-2 composite is the sum of the stress of the single crystals and the polymers, weighted by the volume fraction (v), along the 1, 3, 4, and 5 directions, as shown in Equation (2b). The total strain on the 2-2 composite is the sum of the strain of the single crystals and polymers, weighted by the volume fraction, along the 2 and 6 directions, as shown in Equation (2c). On the other hand, the single crystals and polymers exhibit identical stress along the 2 and 6 directions, as shown in Equation (2d). The single crystals and polymers exhibit identical electric fields along the 1, 2, and 3 directions, as shown in Equation (2e). On the other hand, the total electric displacement on the 2-2 composite is the sum of the electric displacement of the single crystals and the polymers, weighted by the volume fraction, along the 1, 2, and 3 directions, as shown in Equation (2f).
(2a)$$S_{1,3,4,5}^{C}=S_{1,3,4,5}^{P}=S_{1,3,4,5}^{SC}$$
(2b)$$T_{1,3,4,5}^{C}=vT_{1,3,4,5}^{SC}+(1-v)T_{1,3,4,5}^{P}$$
(2c)$$S_{2,6}^{C}=vS_{2,6}^{SC}+(1-v)S_{2,6}^{P}$$
(2d)$$T_{2,6}^{C}=T_{2,6}^{P}=T_{2,6}^{SC}$$
(2e)$$E_{1,2,3}^{C}=E_{1,2,3}^{P}=E_{1,2,3}^{SC}$$
(2f)$$D_{1,2,3}^{C}=vD_{1,2,3}^{SC}+(1-v)D_{1,2,3}^{P}$$

To utilize the above boundary conditions, the driving and environmental constraints for satisfying the boundary conditions should be noted. Since the polymers and single crystals show considerable difference in their compliances, the identical strain boundary condition in Equation (2a) can be violated when the composite drives with excessive high frequency or under excessive hydrostatic pressure. In order to satisfy the identical strain boundary condition of (2a), the aspect ratio of the single crystal pillar and the gap between the single crystal pillar are generally determined by following design rules. The aspect ratio is usually designed to be high enough over 10 and the gap between the pillars is designed to be much less than wavelength at the driving frequency to prevent non-identical strain [42,43]. Furthermore, introducing a shape plate can also constrain the strain of the polymer and single crystals to be identical [44,45]. The boundary conditions (2e) and (2f) may not be satisfied at the edge side of the composite due to the fringe effect, which can lead to local distributions of the electric field and charge field [6,46,47]. This is because of the high difference in the electric displacement coefficients between the single crystal and the polymer. To minimize the influence of the fringe effect, the kerf size can be made sufficiently small, and the electrode surface can be made sufficiently large. In this way, the boundary conditions for the electric field can be satisfied.

The analytic solutions of the equivalent material coefficients of the 2-2 composite are calculated using the constitutive equations 1, 2, and 3 and their boundary conditions 4. The paper includes an appendix with detailed derivations of the analytic solution. The stress equation in directions 1 and 3 is as follows:(3a)$$\begin{array}{ll} T_{i=1,3}^{C} & =\left\{vc_{ij}^{SC}+(1-v)c_{ij}^{P}-\frac{v\left(1-v\right)}{\alpha }\left(c_{i2}^{SC}-c_{i2}^{P}\right)\left(c_{12}^{SC}-c_{12}^{P}\right)\right\}S_{j=1}^{C}+\left\{\frac{1}{\alpha }\left(vc_{ij}^{SC}c_{jj}^{P}+(1-v)c_{jj}^{SC}c_{ij}^{P}\right)\right\}S_{j=2}^{C}\\ & +\left\{vc_{ij}^{SC}+(1-v)c_{ij}^{P}-\frac{v\left(1-v\right)}{\alpha }\left(c_{i2}^{SC}-c_{i2}^{P}\right)\left(c_{23}^{SC}-c_{23}^{P}\right)\right\}S_{j=3}^{C}-\left\{ve_{ij}^{SC}-\frac{v\left(1-v\right)}{\alpha }\left(c_{i2}^{SC}-c_{i2}^{P}\right)e_{2j}^{SC}\right\}E_{j=3}^{C} \end{array}$$
where the coefficient α is as follows:(3b)$$\alpha =vc_{22}^{P}+c_{22}^{SC}(1-v)$$

The stress equation in direction 2 is as follows:(3c)$$T_{i=2}^{C}=\frac{1}{\alpha }\left\{\left(vc_{ij}^{SC}c_{ii}^{P}+(1-v)c_{ii}^{SC}c_{ij}^{P}\right)S_{j=1}^{C}+c_{ij}^{P}c_{ij}^{SC}S_{j=2}^{C}+\left(vc_{ij}^{SC}c_{ii}^{P}+(1-v)c_{ii}^{SC}c_{ij}^{P}\right)S_{j=3}^{C}-\left(ve_{ij}^{SC}c_{ii}^{P}\right)E_{j=3}^{C}\right\}$$

The stress equation in directions 4 and 5 is as follows:(3d)$$T_{i=4,5}^{C}=\left\{\left(vc_{ij}^{SC}+(1-v)c_{ij}^{P}\right)S_{j=4,5}^{C}-\left(ve_{ij}^{SC}\right)E_{j=2,1}^{C}\right\}$$

The stress equation in direction 6 is as follows:(3e)$$T_{i=6}^{C}=\left\{{\left(vs_{ij}^{SC}+(1-v)s_{ij}^{P}\right)}^{-1}\right\}S_{j=6}^{C}$$

The electric displacement equation in directions 1 and 2 is as follows:(3f)$$D_{i=1,2}^{C}=ve_{ij}^{SC}S_{j=5,4}^{C}+\left\{v\varepsilon_{ij}^{SC}+\left(1-v\right)\varepsilon_{ij}^{P}\right\}E_{j=1,2}^{C}$$

The electric displacement equation in direction 3 is as follows:(3g)$$\begin{array}{ll} D_{i=3}^{C} & =\left\{ve_{ij}^{SC}-\frac{v\left(1-v\right)}{\alpha }e_{i2}^{SC}\left(c_{2j}^{SC}-c_{2j}^{P}\right)\right\}S_{j=1}^{C}+\left(\frac{1}{\alpha }ve_{ij}^{SC}c_{jj}^{P}\right)S_{j=2}^{C}\\ & \left\{ve_{ij}^{SC}-\frac{v\left(1-v\right)}{\alpha }e_{2j}^{SC}\left(c_{2j}^{SC}-c_{2j}^{P}\right)\right\}S_{j=3}^{C}+\left\{v\varepsilon_{ij}^{SC}+\left(1-v\right)\varepsilon_{ij}^{P}+\frac{v\left(1-v\right)}{\alpha }e_{i2}^{SC}\right\}E_{j=3}^{C} \end{array}$$

Equations (3a)~(3g) shows the analytic solution of the equivalent material coefficients of the 2-2 composite based on [011] poled single crystals from Equations (4a)~(4q).
(4a)$$c_{11}^{C}=vc_{11}^{SC}+(1-v)c_{11}^{P}-\frac{v\left(1-v\right)}{\alpha }\left(c_{12}^{SC}-c_{12}^{P}\right)\left(c_{12}^{SC}-c_{12}^{P}\right)$$
(4b)$$c_{12}^{C}=\frac{1}{\alpha }\left(vc_{12}^{SC}c_{22}^{P}+(1-v)c_{22}^{SC}c_{12}^{P}\right)$$
(4c)$$c_{13}^{C}=vc_{13}^{SC}+(1-v)c_{13}^{P}-\frac{v\left(1-v\right)}{\alpha }\left(c_{12}^{SC}-c_{12}^{P}\right)\left(c_{23}^{SC}-c_{23}^{P}\right)$$
(4d)$$c_{22}^{C}=\frac{1}{\alpha }c_{22}^{P}c_{22}^{SC}$$
(4e)$$c_{23}^{C}=\frac{1}{\alpha }\left(vc_{23}^{SC}c_{22}^{P}+(1-v)c_{22}^{SC}c_{23}^{P}\right)$$
(4f)$$c_{33}^{C}=vc_{33}^{SC}+(1-v)c_{33}^{P}-\frac{v\left(1-v\right)}{\alpha }\left(c_{32}^{SC}-c_{32}^{P}\right)\left(c_{23}^{SC}-c_{23}^{P}\right)$$
(4g)$$c_{44}^{C}=vc_{44}^{SC}+(1-v)c_{44}^{P}$$
(4h)$$c_{55}^{C}=vc_{55}^{SC}+(1-v)c_{55}^{P}$$
(4i)$$c_{66}^{C}={\left\{vs_{66}^{SC}+(1-v)s_{66}^{P}\right\}}^{-1}$$
(4j)$$e_{13}^{C}=ve_{13}^{SC}-\frac{v\left(1-v\right)}{\alpha }e_{32}^{SC}\left(c_{21}^{SC}-c_{21}^{P}\right)$$
(4k)$$e_{23}^{C}=\frac{1}{\alpha }ve_{23}^{SC}c_{22}^{P}$$
(4l)$$e_{33}^{C}=ve_{33}^{SC}-\frac{v\left(1-v\right)}{\alpha }e_{23}^{SC}(c_{23}^{SC}-c_{23}^{P})$$
(4m)$$e_{24}^{C}=ve_{24}^{SC}$$
(4n)$$e_{15}^{C}=ve_{15}^{SC}$$
(4o)$$\varepsilon_{11}^{C}=v\varepsilon_{11}^{SC}+\left(1-v\right)\varepsilon_{11}^{P}$$
(4p)$$\varepsilon_{22}^{C}=v\varepsilon_{22}^{SC}+\left(1-v\right)\varepsilon_{22}^{P}$$
(4q)$$\varepsilon_{33}^{C}=\varepsilon_{33}^{SC}v+(1-v_{f})\varepsilon_{33}^{P}+\frac{v}{\alpha }\left(1-v\right)e_{23}^{SC}$$

## 3. Finite Element Analysis

The analytic solutions of the equivalent material coefficients derived in Section 2 are verified by finite element analysis. The commercial finite element analysis tool COMSOL Multiphysics with a piezoelectric module with frequency analysis was used for the simulation. The impedance spectrum of single-phase resonators based on the equivalent material coefficients is compared with the impedance spectrum of resonators with a 2-2 configuration. The resonators address more than five mesh elements per wavelength for the accuracy of the simulation.

The 2-2 composite consisting of the [011] poled PIN-PMN-PT single crystals and Epotek 301 epoxy polymer is introduced for simulating the impedance spectrum of resonators. The piezoelectric single crystal [011] poled 0.32Pb(In_1/2_Nb_1/2_)O_3_-0.39Pb(Mg_1/3_Nb_2/3_)O_3_-0.29PbTiO_3_ has been introduced for this study. Table 1 lists material coefficients of [011] poled PIN-PMN-PT grown by the Bridgeman method at iBULe Photonics Inc. The volume fraction of the [011] poled PIN-PMN-PT is assumed to 35%. Table 2 lists the derived equivalent material coefficients of the 2-2 composite by using the analytic solution of Equations (4a)~(4q).

To verify the 17 material coefficients of the 2-2 composites of the [011] poled PIN-PMN-PT single crystals and polymers, 8 resonators are introduced. Each of these resonators are designed to predominantly vibrate in their principal directions, corresponding to specific vibrational modes, as shown in Figure 2 [23]. The name of resonator LE, TE, and LS corresponds to related vibration modes: length extension, thickness extension, and length shear, respectively. The following i and j constants of resonators correspond to the direction of electric field and the direction of dominant deformation, respectively. For the rotated resonator TE31(Z45°), TE31(Y45°), and TE32(X45°), the rotating angle and rotating axis are provided. Resonators are poled in (3) directions, and electrodes are deposited perpendicular to the driving field direction, as shown in Figure 2. Table 3 shows the dimensions of the resonators with their related material coefficients. The dimensions of resonators were determined by referring to the IEEE Standard [24].

Figure 3 shows the results of the impedance spectrum of the single-phase resonators with the equivalent material coefficients and the impedance spectrum of the resonators with 2-2 configuration.

For the LE33, TE31, and TE32 resonators, the impedance curves of the single-phase resonators with equivalent material coefficients and the 2-2 composite resonators are matched well. For the LS15 and LS24 resonators, the impedance curves at resonance frequencies are in relatively good agreement, but the difference in anti-resonance frequency is around 10.05, or 1.36%. The difference in anti-resonance frequency comes from errors in piezoelectric constants $e_{15}$ and $e_{24}$ because the analytic solutions cannot fully describe the shear mode of the composite. For the TE31(Z45°), TE31(Y45°), and TE32(X45°) resonators, the impedance curves of the single-phase resonators with equivalent material coefficients and the 2-2 composite resonators show the difference in resonance frequencies 8.11%, 3.92%, and 0.00%, respectively, and the difference in anti-resonance frequencies 1.96%, 3.48%, and 2.20%, respectively. These differences come from the sum of errors because elastic and piezoelectric coefficients of the rotated resonators are described by a combination of coefficients in various directions.

## 4. Experiments

The 2-2 composite consisting of the [011] poled PIN-PMN-PT single crystals and Epotek 301 epoxy polymers was fabricated as shown in Figure 4. The measured impedance of the fabricated composite was compared to the simulated impedance based on the equivalent material coefficients to verify the analytic solutions. The PIN-PMN-PT single crystals, grown by the Bridgman method, were diced and filled with epoxy to fabricate the 2-2 composite. Epoxy was filled in the diced kerf in a vacuum chamber and was cured at 40 °C room temperature. After lapping and depositing the electrode on the top and bottom surfaces of the composite, the composite was poled by applying the electric field of 5 kV/cm for 30 min. The length, width, and thickness of the fabricated composite are 20 mm, 6 mm, and 2.75 mm, respectively. The volume fraction of the active material in the fabricated composite is 0.35, and the kerf size (the width of the polymer) is minimized to prevent pillar interaction between single crystals and the polymer, with a size of 500 μm. The electrode plate of the composite was attached on the top and bottom surface of the composites.

Figure 5 shows the measured impedance spectrum of the 2-2 composite and the simulated impedance curves of the single-phase resonator based on the equivalent material coefficients. The impedance spectrums were obtained by measuring the driving voltage and current with respect to frequency using an impedance analyzer (E4990A/KEYSIGHT). The measured and simulated impedance curves are well matched in the thickness vibration mode. The measured resonance frequency and anti-resonance frequency of the thickness mode were 436 kHz and 560 kHz, respectively. The simulated resonance frequency and anti-resonance frequency of the thickness mode were 435 kHz and 553 kHz, respectively. The difference in resonance and anti-resonance frequencies of the thickness mode is around 0.5% and 1.2%. The other resonance frequencies around 280 kHz and 360 kHz correspond to the resonance frequencies of the lateral vibration mode. The results verify the analytic solution for the equivalent material coefficients. 

## 5. Conclusions

In this study, the analytic solution for equivalent material coefficients of the 2-2 is introduced based on composite constitutive equations and their boundary conditions. The analytic solutions and the derived equivalent material coefficients are verified by finite element analysis. The eight resonators, of which the impedance spectrums are related to the material coefficients, are introduced. The results show that the derived material coefficients related to the length deformation are quite accurate and the derived material coefficients related to the shear deformation show relatively larger errors. Furthermore, the accuracy of the derived equivalent material coefficients utilized in the analytic solution was confirmed by using the measured impedance spectrum of the fabricated 2-2 composite. The difference in resonance and anti-resonance frequencies between the measured and simulated impedance spectrums was around 0.5% and 1.2%, respectively. The 2-2 composite plate utilizing [011] poled PIN-PMN-PT and Epotek 301 epoxy is fabricated. The measured impedance curves of the fabricated composites are well matched to the simulated curves with the equivalent material properties. Even though the analytic solution does not address the errors from local distribution of mechanical and electric fields, the analytic solution sufficiently demonstrates primary characteristics of the 2-2 composites with minimal errors. This analytic solution of the 2-2 composite provides great advantage over other methods to identify material properties such as the testing method or numerical simulation. Furthermore, since 2-2 composites based on a [011] poled single crystal provide superior properties for acoustic sensors, the analytic solution for the material properties of the composite is essential for designing acoustic sensors.

## Figures and Tables

**Figure 1 micromachines-15-00917-f001:**
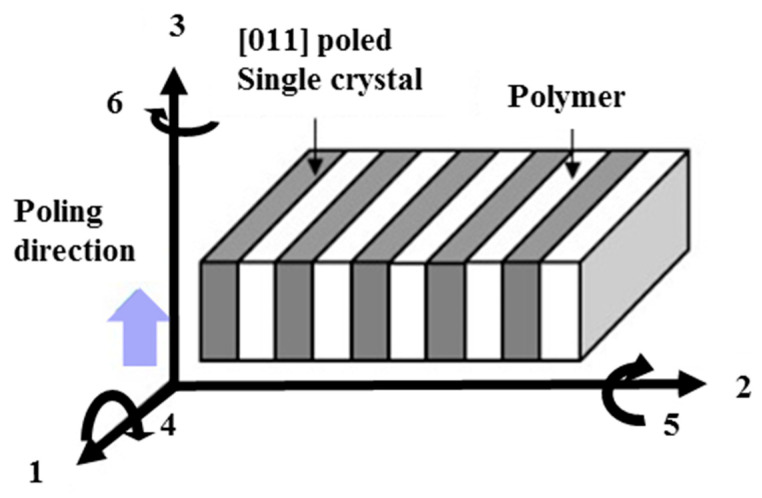
Schematic diagram of 2-2 composites consisting of the [011] poled single crystals and polymers with the polling directions in the (3) directions: (**1**) normal component along the *x*-axis, (**2**) normal component along the *y*-axis, (**3**) normal component along the z-axis, (**4**) shear component in the x-y plane, (**5**) shear component in the y-z plane, (**6**) shear component in the z-x plane.

**Figure 2 micromachines-15-00917-f002:**
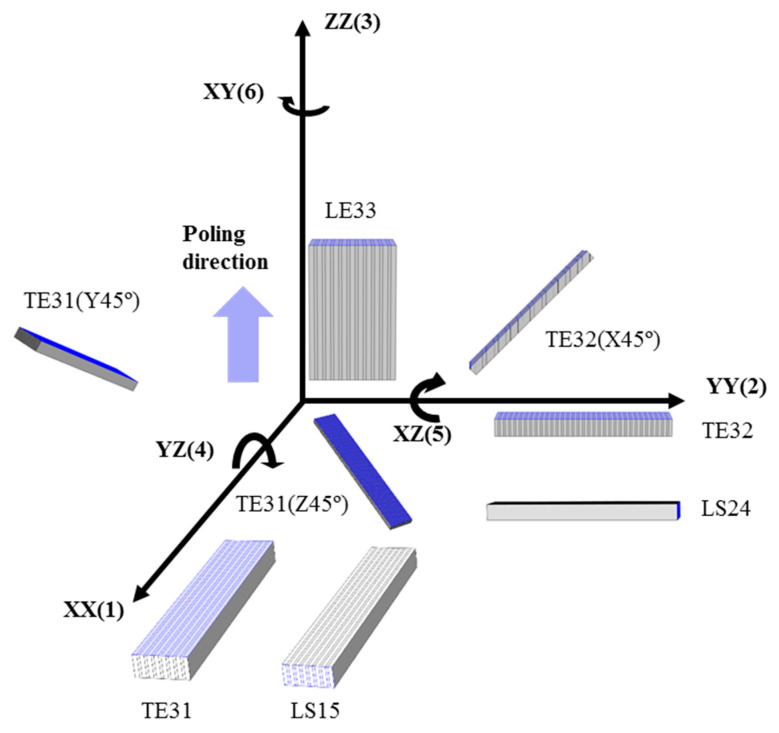
Schematic drawing of 2-2 composites consisting of the [011] poled PIN-PMN-PT single crystals and polymers.

**Figure 3 micromachines-15-00917-f003:**
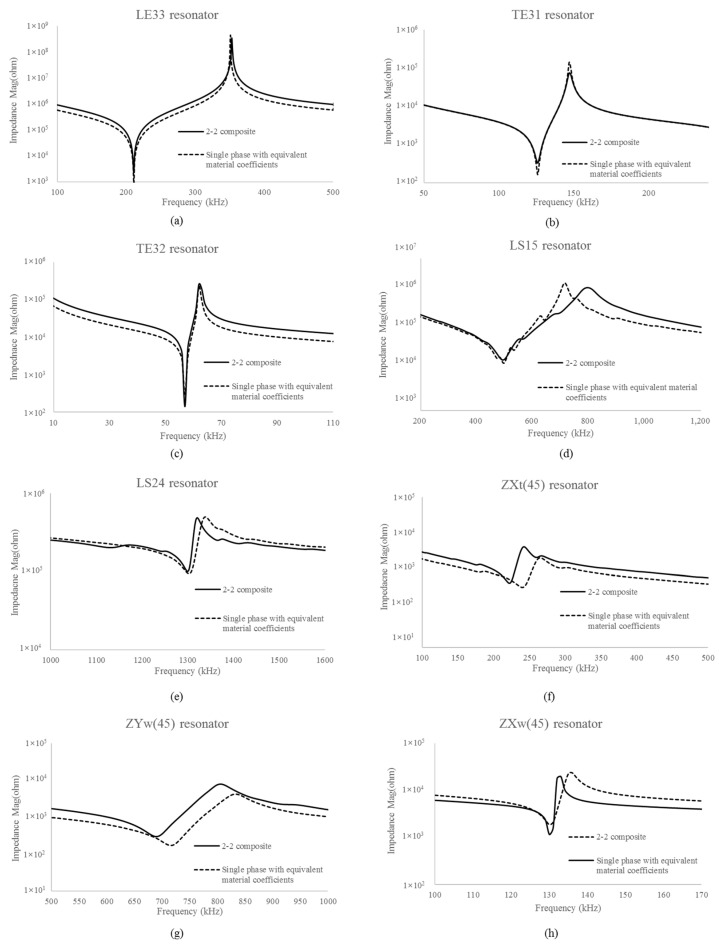
Comparison of impedance spectrum of single-phase resonator with equivalent material coefficient resonators and 2-2 composite resonators: (**a**) LE33 resonator, (**b**) TE31 resonator, (**c**) TE32 resonator, (**d**) LS15 resonator, (**e**) LS24 resonator, (**f**) TE31(Z45°) resonator, (**g**) TE31(Y45°) resonator, and (**h**) TE32(X45°) resonator.

**Figure 4 micromachines-15-00917-f004:**
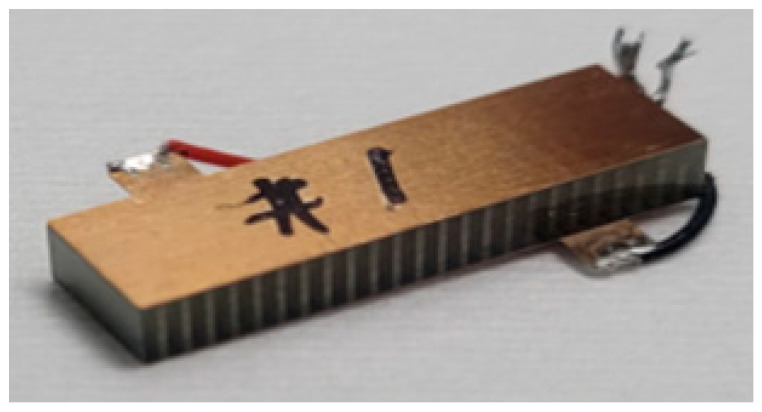
Fabricated sample of 2-2 composite consisting of [011]-poled PIN-PMN-PT single crystals and Epotek 301 epoxy polymers.

**Figure 5 micromachines-15-00917-f005:**
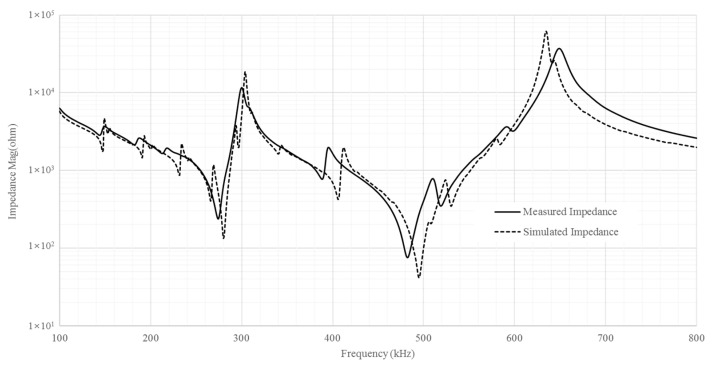
Measured (solid line) and calculated (dashed line) impedance spectrums.

**Table 1 micromachines-15-00917-t001:** Material coefficients of the [011] poled PIN-PMN-PT single crystal and Epotek 301 epoxy polymer.

[011] Poled PIN-PMN-PT								
$c_{11}$ (GN/m^2^)	$c_{12}$ (GN/m^2^)	$c_{13}$ (GN/m^2^)	$c_{22}$ (GN/m^2^)	$c_{23}$ (GN/m^2^)	$c_{33}$ (GN/m^2^)	$c_{44}$ (GN/m^2^)	$c_{55}$ (GN/m^2^)	$s_{66}$ (pm^2^/N)
225.02	54.254	−44.44	91.176	82.54	152.51	67.48	7.84	13.80
$e_{31}$ (C/m^2^)	$e_{32}$ (C/m^2^)	$e_{33}$ (C/m^2^)	$e_{15}$ (C/m^2^)	$e_{24}$ (C/m^2^)	$\varepsilon_{11}$ (PC/ m^2^)	$\varepsilon_{22}$ (PC/ m^2^)	$\varepsilon_{33}$ (PC/ m^2^)	ρ (kg/m^3^)
−10.49	−13.42	13.61	17.29	6.01	849.2	1019.4	508.2	8160
Epotek 301 epoxy polymer								
$c_{11}$ (GN/m^2^)	$c_{12}$ (GN/m^2^)	$c_{13}$ (GN/m^2^)	$c_{22}$ (GN/m^2^)	$c_{23}$ (GN/m^2^)	$c_{33}$ (GN/m^2^)	$c_{44}$ (GN/m^2^)	$c_{55}$ (GN/m^2^)	$s_{66}$ (pm^2^/N)
4.30	2.12	2.12	4.30	2.12	4.30	1.09	1.09	458.62
$\varepsilon_{11}$ (PC/ m^2^)		$\varepsilon_{22}$ (PC/ m^2^)			$\varepsilon_{33}$ (PC/ m^2^)		ρ (kg/m^3^)	
4		4			4		1150	

**Table 2 micromachines-15-00917-t002:** Equivalent material coefficients of the 2-2 composite.

Equivalent Material Coefficients of 2-2 Composite								
$c_{11}$ (GN/m^2^)	$c_{12}$ (GN/m^2^)	$c_{13}$ (GN/m^2^)	$c_{22}$ (GN/m^2^)	$c_{23}$ (GN/m^2^)	$c_{33}$ (GN/m^2^)	$c_{44}$ (GN/m^2^)	$c_{55}$ (GN/m^2^)	$c_{66}$ (GN/m^2^)
71.37	3.47	−29.88	6.45	4.11	31.96	24.33	3.45	3.30
$e_{31}$ (C/m^2^)	$e_{32}$ (C/m^2^)	$e_{33}$ (C/m^2^)	$e_{15}$ (C/m^2^)	$e_{24}$ (C/m^2^)	$\varepsilon_{11}$ (PC/m^2^)	$\varepsilon_{22}$ (PC/m^2^)	$\varepsilon_{33}$ (PC/m^2^)	ρ (kg/m^3^)
−1.05	−0.33	11.0	6.05	2.10	609.4	360.0	418.1	3604

**Table 3 micromachines-15-00917-t003:** Dimensions of the resonators and their related material coefficients of the resonators for property measurement of the 2-2 composite.

Resonators	Dimensions (X × Y × Z, mm)	Related Material Coefficients
LE33	1 × 1 × 5	$e_{33}$, $c_{33}$
TE31	12 × 2 × 1	$e_{31}$, $c_{11}$
TE32	13 × 2 × 1	$e_{32}$, $c_{22}$
LS15	2 × 10 × 1	$e_{15}$, $c_{55}$
LS24	2 × 12 × 1	$e_{24}$, $c_{44}$
TE31(Z45°)	12 × 2 × 0.4	$c_{11}$, $c_{22}$, $c_{12}$, $c_{66}$
TE31(Y45°)	7 × 2 × 0.4	$c_{22}$, $c_{33}$, $c_{23}$, $c_{44}$
TE32(X45°)	6 × 1.2 × 0.4	$c_{11}$, $c_{33}$, $c_{13}$, $c_{55}$

## Data Availability

Data are contained within the article.

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
