# Peer review of "Derivation of Equivalent Material Coefficients of 2-2 Piezoelectric Single Crystal Composite"

_micromachines, 2024, doi:10.3390/mi15070917_

Round 1

Reviewer 1 Report

Comments and Suggestions for Authors

Revision of “Derivation of equivalent material coefficients of 2-2 piezoelectric single crystal composite”

The manuscript under review is devoted to the investigation of an analytical solution for the determination of equivalent material coefficients of the 2-2 composite based on constitutive equations and their boundary conditions. The analytic solutions and the derived equivalent material coefficients were verified by Finite Element Analysis. Providing of such investigations is very important from an academic and economic points of view.

The 8 resonators, of which impedance spectrums are related to the material coefficients were introduced. It was shown that the derived material coefficients related to the length direction are quite accurate and the derived material coefficients related to the shear direction show relatively larger errors. Furthermore, the accuracy of the derived equivalent material coefficients utilized in the analytic solution was confirmed by using the measured impedance spectrum of the fabricated 2-2 composite. The difference in resonance and anti-resonance frequencies between the measured and simulated impedance spectrums was around 0.5% and 1.2%, respectively.

In manuscript all necessary information is captured by 5 figures and 3 tables. There are 28 references, all of them are reflected in the text.

After getting acquainted with the presented manuscript, a there are few questions remained:

1.      The authors must clearly state what was done for the first time.

2.      If the authors use a single crystal obtained by the Bridgman method, then it is either necessary to indicate the manufacturer who guarantees the properties given in the article, or to provide the production modes.

3.      Why is the specific composition of the single crystal not indicated, since this may change the electrical parameters.

4.      Page 1 line 40 “PbZrTiO3” it is necessary to use the subscript for 3.

5.      Page 8 line 191 it is necessary to remove the dot “figure (2). [23]. The...”

The obtained results are important for understanding the physical processes that occur in real objects. The described manuscript is corresponds to the field of the Journal. It may be accepted after revision.

Author Response

Authors Reply to Reviewer 1

The authors appreciate the kind and detailed comments of the reviewer. The manuscript entitled " Derivation of equivalent material coefficients of 2-2 piezoelectric single crystal composite" has been substantially revised according to the comments of the reviewer. The revision has been made as follows.

1.   The authors must clearly state what was done for the first time.

-We revised the introduction to clearly highlight our novel contributions, addressing the reviewer's comments. We checked all the references in the manuscript and add some new reference of recent studies taking account the reviewer’s comment. Please see the 5th paragraph of Section 1 of the revised manuscript.

2.   If the authors use a single crystal obtained by the Bridgman method, then it is either necessary to indicate the manufacturer who guarantees the properties given in the article, or to provide the production modes.

-We revised the manuscript to indicate the manufacturer of the single crystal. We also rearranged sentences and paragraphs for better understanding in the revised manuscript. Please see the 3rd paragraph of Section 3 of the revised manuscript.

3.   Why is the specific composition of the single crystal not indicated, since this may change the electrical parameters. 

-We revised the manuscript to specific the composition of the single crystal. Please see the 3rd paragraph of Section 3 of the revised manuscript. 

4.   Page 1 line 40 “PbZrTiO3” it is necessary to use the subscript for 3

We corrected typo errors. Please see the changes in the revised manuscript.

5.   Page 8 line 191 it is necessary to remove the dot “figure (2). [23]. The...”

We corrected typo errors. Please see the changes in the revised manuscript.

Reviewer 2 Report

Comments and Suggestions for Authors

Dear Authors,

Your work is devoted to a current problem. The research results can be used to solve various acoustic problems. Publication of the article seems appropriate.

 Some comments on the text of the work:

1. The Authors calculate a number of parameters 2-2 piezoelectric single crystal composite using the finite element method. However, the progress of this decision is not given in the article. The process of numerical solution should be described in more detail at the beginning of Section 3 (software used, solution progress, mesh dimensions, etc.).

2. It is necessary to describe in more detail the process of obtaining the experimental data presented in Fig. 5. The equivalent circuit and the equipment used to determine the resistance of the 2-2 piezoelectric single crystal composite depending on the frequency are also desirable.

3. Lines 27-29: “Piezoelectric composites, which consist of piezoelectric materials and polymers, are widely employed in various applications such as underwater sonar transducers, medical diagnostic ultrasonic transducers, and non-destructive testing devices [1-3].» The application of the solutions proposed by the authors in such wide areas must take into account the complex nature of the physical connections between the components of the composite.

Since the article talks about static and quasi-static measurements of parameters, the question arises about the influence of dynamic loads, temperature, pressure, as well as technological parameters (for example, the thickness of the contact layer between the electrode and the composite, if any) on the parameters of the composite.

It is necessary to clarify the limits of applicability of the proposed design solutions for natural conditions.

         4. Spotted typos: Lines 165-167 – present twice in the numbering of formulas (5d).

Author Response

Authors Reply to Reviewer 2

The authors appreciate the kind and detailed comments of the reviewer. The manuscript entitled " Derivation of equivalent material coefficients of 2-2 piezoelectric single crystal composite" has been substantially revised according to the comments of the reviewer. The revision has been made as follows.

1.   The Authors calculate a number of parameters 2-2 piezoelectric single crystal composite using the finite element method. However, the progress of this decision is not given in the article. The process of numerical solution should be described in more detail at the beginning of Section 3 (software used, solution progress, mesh dimensions, etc.)

We revised the manuscript to describe more detains in finite element method. Please see the 6th paragraph of Section 3 of the revised manuscript. 

2.   It is necessary to describe in more detail the process of obtaining the experimental data presented in Fig. 5. The equivalent circuit and the equipment used to determine the resistance of the 2-2 piezoelectric single crystal composite depending on the frequency are also desirable.

We revised the manuscript to describe in more detail process of obtaining the experimental data. Please see the 1st and 2nd of the Section 4 of the revised manuscript.

3.   Lines 27-29: “Piezoelectric composites, which consist of piezoelectric materials and polymers, are widely employed in various applications such as underwater sonar transducers, medical diagnostic ultrasonic transducers, and non-destructive testing devices [1-3].» The application of the solutions proposed by the authors in such wide areas must take into account the complex nature of the physical connections between the components of the composite. Since the article talks about static and quasi-static measurements of parameters, the question arises about the influence of dynamic loads, temperature, pressure, as well as technological parameters (for example, the thickness of the contact layer between the electrode and the composite, if any) on the parameters of the composite. 

We revised the introduction to explain the complex nature of physical connections between the components of the composite. We checked all the references in the manuscript and add some new reference of recent studies taking account the reviewer’s comment. Please see the 5th paragraph of Section 1 of the revised manuscript.

4.   Spotted typos: Lines 165-167 – present twice in the numbering of formulas (5d).

We corrected typo errors. Please see the changes in the revised manuscript. 

Reviewer 3 Report

Comments and Suggestions for Authors

In introduction, the state-of-the-art literature is not elaborated enough, and the novelty factor of the study is not identified, considering that analytical solutions for configurations 2-2 piezoelectric-polymer compositions have already been made.

The major concern is related to the boundary conditions introduced in study, especially for the strain (“single crystals and polymers exhibit identical strain”) and electric field (“The single crystals and polymers exhibit identical electric fields”). If the analysis is based on the wrong conditions, it may involve erroneous reasoning and results.

The aspect related to the very high difference between the permittivity value of the polymer (below 10) and the crystal (higher than 500) that will considerably change the way the local electric field is distributed is not taken into account at all (there are already reported studies on this subject!).

The experimental part is not convincing, with an insufficient description of the equipment used for measurements, the poling condition for the composite system to measure resonance and anti-resonance, and the dependence on the dimensions of the composite material.

The conclusions can be improved by highlighting the novelty factor

Minor text editing is requested (for example: PbZrTiO3 (PZT)-3 must be as subscript). 

The article can be accepted only after a major revision.

Author Response

Author’s Reply to Reviewer 3

The authors appreciate the kind and detailed comments of the reviewer. The manuscript entitled " Derivation of equivalent material coefficients of 2-2 piezoelectric single crystal composite" has been substantially revised according to the comments of the reviewer. The revision has been made as follows.

  1. In introduction, the state-of-the-art literature is not elaborated enough, and the novelty factor of the study is not identified, considering that analytical solutions for configurations 2-2 piezoelectric-polymer compositions have already been made.
  • We revised the introduction to clearly highlight our novel contributions, addressing the reviewer's comments. We checked all the references in the manuscript and add some new reference of recent studies taking account the reviewer’s comment. Please see the 5th paragraph of Section 1 of the revised manuscript.
  1. The major concern is related to the boundary conditions introduced in study, especially for the strain (“single crystals and polymers exhibit identical strain”) and electric field (“The single crystals and polymers exhibit identical electric fields”). If the analysis is based on the wrong conditions, it may involve erroneous reasoning and results. The aspect related to the very high difference between the permittivity value of the polymer (below 10) and the crystal (higher than 500) that will considerably change the way the local electric field is distributed is not taken into account at all (there are already reported studies on this subject!).
  • We revised the manuscript to describe the constraints for utilizing the boundary conditions related to the strain and the electric field. The application area of our theoretical model is also stated. Theoretical model considering the effect of temperature change, hydrostatic pressure, and high-frequency driving will be investigated in further study. Please see the 7th paragraph of the Section 2 of the revised manuscript
  1. The experimental part is not convincing, with an insufficient description of the equipment used for measurements, the poling condition for the composite system to measure resonance and anti-resonance, and the dependence on the dimensions of the composite material.
  • We revised the manuscript to describe in more detail process of obtaining the experimental data. Please see the Section 4 of the revised manuscript.
  1. The conclusions can be improved by highlighting the novelty factor
  • We revised the manuscript to highlight the novelty factor in conclusions. Please see the changes in the revised manuscript.
  1. Minor text editing is requested (for example: PbZrTiO3 (PZT)-3 must be as subscript).
  • Please see the changes in the revised manuscript.

Round 2

Reviewer 3 Report

Comments and Suggestions for Authors

In the introduction for the novelty aspect, the authors introduce the following paragraph: „However, there has been no study investigating the theoretical model of the 2-2 composites based on [011] poled single crystal having strong anisotropy”.  However, after a very short search, I found an article and a review (https://doi.org/10.1039/D1CE01455B; https://doi.org/10.1016/j.pmatsci.2014.10.002) that shows in the literature are similar studies carried out by other authors on this topic. The authors must reformulate the aspect of novelty (probably related only to the materials used in the current study) and update the references.

I don't understand how the simulations remained unchanged after the introduction of different/new conditions (eq. 5e-g). This aspect is a big question mark. The authors did not discuss anything about this aspect in their response. The problem of the distribution of local fields was not solved, and it was not even mentioned that the present study does not involve this aspect.

Therefore, a major revision is still needed. 

Author Response

Please see the file titled "Author reply to Reviewer". Thank you

Round 3

Reviewer 3 Report

Comments and Suggestions for Authors

The authors failed to understand and respond clearly to the reviewer's requests. However, I believe the manuscript can move forward after minor changes, as it presents interesting elements that deserve publication. It is desirable that the authors include some comments in the manuscript regarding the limitations of the presented study and its potential as a base for future research.

Author Response

Author’s Reply to Reviewer

The authors appreciate the kind and detailed comments of the reviewer. The manuscript entitled " Derivation of equivalent material coefficients of 2-2 piezoelectric single crystal composite" has been substantially revised according to the comments of the reviewer. The revision has been made as follows.

  1. The authors failed to understand and respond clearly to the reviewer's requests. However, I believe the manuscript can move forward after minor changes, as it presents interesting elements that deserve publication. It is desirable that the authors include some comments in the manuscript regarding the limitations of the presented study and its potential as a base for future research.
  • We revised the manuscript to explain the limitations of the study. Please see Section 2 and the conclusions of the revised manuscript.